# Simulating hyperbolic space on a circuit board

Patrick M. Lenggenhager[1,2,3,9], Alexander Stegmaier[4,9], Lavi K. Upreti [4], Tobias Hofmann [4], Tobias Helbig [4], Achim Vollhardt [2], Martin Greiter [4], Ching Hua Lee [5], Stefan Imhof[6], Hauke Brand [6], Tobias Kießling[6], Igor Boettcher[7,8], Titus Neupert [2] ✉, Ronny Thomale [4] ✉ & Tomáš Bzdušek[1,2] ✉

The Laplace operator encodes the behavior of physical systems at vastly different scales, describing heat flow, fluids, as well as electric, gravitational, and quantum fields. A key input for the Laplace equation is the curvature of space. Here we discuss and experimentally demonstrate that the spectral ordering of Laplacian eigenstates for hyperbolic (negatively curved) and flat two-dimensional spaces has a universally different structure. We use a lattice regularization of hyperbolic space in an electric-circuit network to measure the eigenstates of a 'hyperbolic drum', and in a time-resolved experiment we verify signal propagation along the curved geodesics. Our experiments showcase both a versatile platform to emulate hyperbolic lattices in tabletop experiments, and a set of methods to verify the effective hyperbolic metric in this and other platforms. The presented techniques can be utilized to explore novel aspects of both classical and quantum dynamics in negatively curved spaces, and to realise the emerging models of topological hyperbolic matter.

Curved spaces, traditionally studied in high-energy physics and cosmology, have recently been elevated to paramount importance in condensed matter physics for two reasons. First, the discovery of holographic principles[1,2] revealed a fundamental hidden structure underlying certain interacting quantum many-body systems that allows to compute their properties from a theory in hyperbolic space of negative curvature. Remarkably, these insights have been applied successfully to analyze strongly correlated electronic systems with tools from holography and to gain insight into the nature of quantum entanglement in condensed matter systems[3–11]. Second, major advancements in the mathematical characterization of classical and quantum states in negatively curved spaces[12–15] sparked a resurgence of interest of the condensed matter and metamaterials communities in hyperbolic lattices[16–18], ushering the research of hyperbolic topological matter[19–21]. These rapid developments call for new experimental platforms to implement tabletop simulations of hyperbolic toy-models.

However, systems that furnish negatively curved space[22,23] are hard to realize experimentally. The mathematical reason for this is encompassed in Hilbert's theorem: even the lowest dimensional model of a hyperbolic space, the hyperbolic plane, cannot be embedded in three-dimensional Euclidean (flat) laboratory space. We cannot build a hyperbolic drum. This is in sharp contrast to the case of positive curvature: a sphere can be embedded in three-dimensional space, and we can study the standing waves (hereafter called eigenmodes) of a spherical membrane, which directly relate to quantum numbers of atomic orbitals. Despite such obstacles, hyperbolic space can be

[1]Condensed Matter Theory Group, Paul Scherrer Institute, 5232 Villigen PSI, Switzerland. [2]Department of Physics, University of Zurich, Winterthurerstrasse 190, 8057 Zurich, Switzerland. [3]Institute for Theoretical Physics, ETH Zurich, 8093 Zurich, Switzerland. [4]Institut für Theoretische Physik und Astrophysik, Universität Würzburg, 97074 Würzburg, Germany. [5]Department of Physics, National University of Singapore, Singapore 117551, Republic of Singapore. [6]Physikalisches Institut, Universität Würzburg, 97074 Würzburg, Germany. [7]Department of Physics, University of Alberta, Edmonton, AB T6G 2E1, Canada. [8]Theoretical Physics Institute, University of Alberta, Edmonton, AB T6G 2E1, Canada. [9]These authors contributed equally: Patrick M. Lenggenhager, Alexander Stegmaier. ✉e-mail: titus.neupert@uzh.ch; rthomale@physik.uniwuerzburg.de; tomas.bzdusek@psi.ch

emulated experimentally. For instance, it has been suggested[24] that a non-trivial metric can be implemented in metamaterials via spatial variations of the electromagnetic permittivity of continuous media. However, it is very challenging to induce these variations in a controlled manner, which limits the applicability of such approaches.

Electric circuits[25–32] and similar systems, e.g., coplanar waveguide resonators[16], overcome these experimental limitations by relying on a discretization of space. In electric circuit networks, the physical distances between the nodes are fundamentally decoupled from the metric that enters the long-wavelength description of its degrees of freedom, namely the voltages and currents that pass through the circuit nodes. The latter depend merely on the circuit elements that connect the nodes. Compared to other experimental platforms, electric circuits significantly excel in their flexibility of design, ease of fabrication, and high accessibility to measurements.

In this work we present a strategy for verifying that electric circuits can emulate the physics of negatively curved spaces and we demonstrate that electric circuits can do so efficiently. For concreteness, we consider the most fundamental differential operator on curved spaces, the Laplace-Beltrami operator, which generalizes the notion of the Laplace operator on flat space. The first key result of our work is the experimental observation of negative curvature in the spectral ordering of the eigenmodes of the Laplace-Beltrami operator in hyperbolic space. To paraphrase the words of ref. 33, our measurements confirm that a hyperbolic drum has a sound distinct from a Euclidean drum. Second, since electric circuits allow for time-resolved measurements, we can study not only static, but also dynamic properties. Our measurements confirm that signals in the present realization travel along hyperbolic geodesics, a smoking gun signature for the negative curvature of space. Based on our results, we infer that electric circuit networks could be readily utilized to implement and to experimentally verify the predicted features of the recently studied hyperbolic models of refs. 13–20. We expect the presented methodology for extracting fingerprints of negative curvature to be generalizable to other platforms, in particular to superconducting waveguide resonators that may allow for exciting future incorporation of quantum phenomena[16].

## Results

### Spectra of Euclidean and hyperbolic drums

We start by comparing the eigenmodes of Euclidean and hyperbolic drums in the continuum. The hyperbolic plane, characterized by a constant negative Gaussian curvature $K < 0$, is naturally embedded in $(2 + 1)$-dimensional Minkowski space as a hyperboloid sheet with fixed timelike distance from the origin, see Fig. 1a. To solve for the eigenmodes of the wave equation, it is convenient to set $K = -4$ and to employ the stereographic projection Fig. 1a, which maps the

hyperbolic plane onto the Poincaré disk, i.e., the unit disk with length element $\mathrm{d}s^2 = (1 - x^2 - y^2)^{-2}(\mathrm{d}x^2 + \mathrm{d}y^2)$.

The eigenmodes of the hyperbolic drum with $x^2 + y^2 \leq r_0^2 < 1$ correspond to the spectrum[17,34,35] of the Laplace-Beltrami operator:

$$\Delta_H = (1 - x^2 - y^2)^2 \Delta_E, \qquad (1)$$

where $\Delta_E = (\partial^2/\partial x^2 + \partial^2/\partial y^2)$ is the usual Laplace operator in the Euclidean plane. Adopting Dirichlet boundary conditions, which yield vanishing amplitude on the disk boundary, the spectrum of the drum is given by solutions to:

$$-\Delta_g u_g^{n\ell} = \lambda_g^{n\ell} u_g^{n\ell} \quad \text{with} \quad u_g^{n\ell}|_{x^2 + y^2 = r_0^2} = 0, \qquad (2)$$

where $g \in \{E, H\}$ indicates the geometry, and $\lambda_g^{n\ell}$ is the frequency of the mode with angular momentum $\ell$ and with $n$ radial zeroes. Solutions to Eq. (2) are superpositions of Bessel functions (associated Legendre functions) in the Euclidean (hyperbolic) case, cf. Methods.

We plot in Fig. 1b the first few solutions to Eq. (2) on the Euclidean vs. Poincaré disk for $r_0 = 0.94$, which corresponds to our experimental realization discussed below. We observe a significant reordering of the eigenmodes characterized by $(n, \ell)$: while in the Euclidean case the first eigenmode with $n = 1$ is the fourth (not counting degenerate eigenmodes separately), in the hyperbolic case, it is only the sixth mode. This reordering becomes even more apparent when considering the angular momentum dispersion $\lambda_g^{n\ell}$ vs. $\ell$ displayed in Fig. 2a. In both the Euclidean and the hyperbolic case, several branches (corresponding to different values of $n$, indicated by red numbers) are discernible. The spectral reordering manifests as a reduced slope of the branches (relative to their spacing) compared to their behavior for the Euclidean drum. Consequently, eigenmodes with large $\ell$ and small $n$ appear much earlier in the spectrum in hyperbolic compared to Euclidean space. The spectral reordering is stronger for larger radii $r_0$. This is intuitively understood from the fact that the circumference of a hyperbolic drum grows superlinearly with its radius, such that oscillations in the angular direction stretch over larger distances. This makes them energetically favorable over oscillations in the radial direction, resulting in the observed reordering.

### Lattice regularization of the hyperbolic plane

To experimentally realize a hyperbolic drum in an electric circuit network, we discretize the continuous space formed by the hyperbolic plane. This is achieved by tessellating the hyperbolic plane with regular polygons; a regular tessellation with $q$ copies of $p$-sided polygons meeting at each vertex is conventionally denoted by the Schläfli

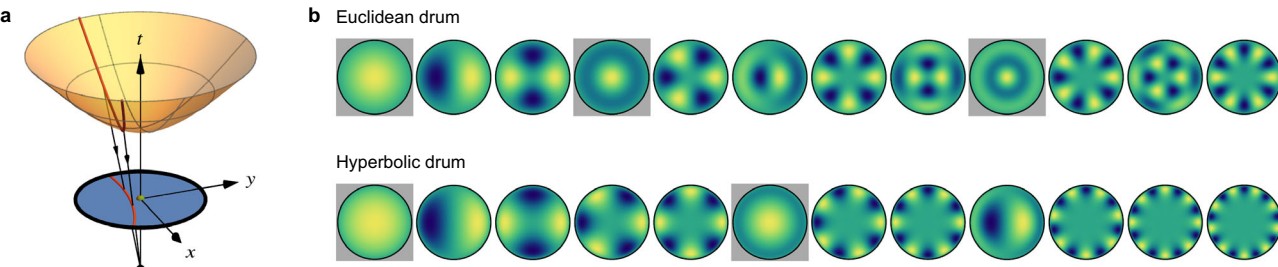

**a** Euclidean drum

Hyperbolic drum

**Fig. 1 | Continuum spectra. a** The hyperboloid (orange) defined by $t^2 - x^2 - y^2 = +1$ in $(2 + 1)$-dimensional $(x, y, t)$ Minkowski space is mapped (black rays) by the stereographic projection through the point $(0, 0, -1)$ (black dot) to the unit disk (blue) at $t = 0$. The geodesics (red) are given by intersections of the hyperboloid with planes passing through the origin $(0, 0, 0)$ (green dot), and are mapped by the projection to circular arcs perpendicular to the boundary of the Poincaré disk.

**b** Comparison of the first few eigenmodes of the Euclidean and hyperbolic drum of radius $r_0 = 0.94$ according to increasing eigenvalues $\lambda_g^{n\ell}$. Their spatial profile $u_g^{n\ell}$ is shown with yellow (green, blue) denoting maxima (zeros, minima). The number of radial zeros inside the disk, $n$, and the angular momentum (number of angular zeros), $\ell$, can easily be inferred from the plots. Modes with $\ell = 0$ are indicated with a gray background.

symbol $\{p, q\}$. The curvature of the continuous space constrains the possible choices of $p$ and $q$: for vanishing curvature (Euclidean plane) they need to satisfy $(p-2)(q-2) = 4$, while negative curvature (hyperbolic plane) requires $(p-2)(q-2) > 4$. A given regular hyperbolic tessellation uniquely fixes the distance between neighboring sites (cf. Supplementary Note 3), in contrast to the Euclidean case where the distance can be scaled arbitrarily.

Interpreting the vertices as sites of a lattice and the edges as connections between nearest neighbors, we obtain a hyperbolic lattice. The sites and nearest-neighbor connections form a graph whose Laplacian matrix gives the lattice regularization of the continuum Laplace-Beltrami operator[17], which is fully determined by the topology of the lattice. The metric of the underlying continuous space is manifested in the connectivity of the lattice sites and therefore in the graph without reference to the positions of the vertices. However, the positions of the graph nodes (i.e., lattice sites) are relevant for the interpretation of the graph as a lattice when explaining the effective physics.

Different tessellations of the hyperbolic plane are possible, and they generally differ in their symmetries and in how densely their vertices cover the disk. For our experiments, three different aspects of the modeled lattice are important: (i) the lattice should provide a good approximation of the continuum, (ii) a large fraction of the Poincaré disk should be covered to obtain strong signatures of the negative curvature, and (iii) $\ell = 0$ modes should be easy to excite and distinguish from $\ell \neq 0$ modes. While aspects (i) and (ii) can both be satisfied by having a sufficiently large number of vertices, in practice, there will be a trade off between the two aspects: for a fixed number of vertices, tessellations with larger area per vertex cover a larger fraction $r_0$ of the Poincaré disk, while for fixed coverage $r_0$ a good approximation of the continuum is naturally achieved by tessellations that feature small area per vertex (i.e., which tile the hyperbolic plane densely)[17]. Finally, (iii) depends on the symmetry properties of the lattice: a vertex at the origin of the disk allows for easy excitation and identification of $\ell = 0$ modes and a high order of rotation symmetry prevents $\ell \neq 0$ modes to have non-vanishing weight at the origin of the disk, which would impede the identification of $\ell = 0$ modes. We analyze and compare several different tessellations with respect to these three aspects in the Supplementary Note 3. These considerations favor the $\{3, 7\}$ tessellation, which exhibits a seven-fold rotation symmetry with respect to a site at the center, and which covers a disk with radius $r_0 = 0.94$ with only 85 sites, see Fig. 3a.

In the long-wavelength-regime, eigenvectors of the Laplacian matrix can be associated with eigenmodes of the Laplace-Beltrami operator in the continuum. We match them by systematically determining the absolute value of the angular momentum $\ell$ of the eigenvectors by a Fourier transform of their components on the outermost sites. Due to the discreteness of the lattice, this analysis is only reliable for modes with sufficiently small $\ell$ and $n$, i.e., in the long-wavelength limit. Note that while the Laplacian matrix is defined purely on the graph, to define angular momentum we need to interpret the graph as a regular lattice, i.e., identify the vertices with lattice sites. But, as mentioned above, the (relative) positions of those sites are uniquely defined by the graph via the values of $p$ and $q$. We extract the angular momentum dispersion for the chosen tessellation, and in Fig. 2b compare it to the corresponding Euclidean $\{3, 6\}$ tessellation with the same number of sites. As in the continuum, a strong spectral reordering is observed. This reordering is a universal feature of the spatial curvature and does, therefore, not rely on the details of the tessellation, as long as it adequately approximates the continuum.

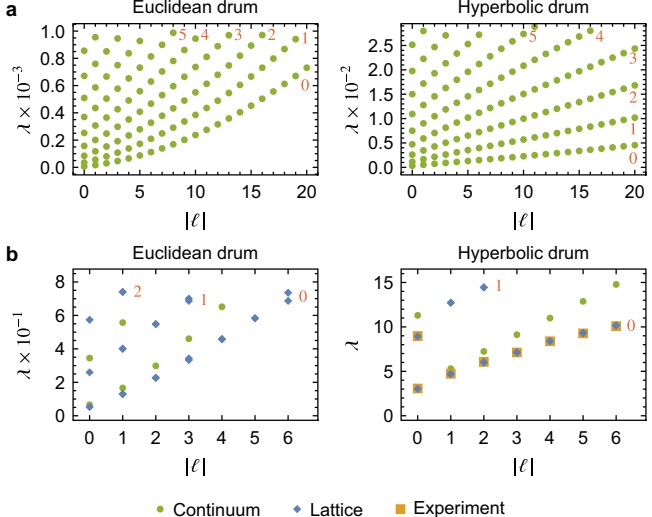

**Fig. 2 | Angular momentum dispersion. a** Rescaled frequency $\lambda_g^{n\ell}$ vs. angular momentum $\ell$ for eigenmodes $u_g^{n\ell}$ of the continuum Laplace-Beltrami operator, i.e., solutions to Eq. (2), for the Euclidean (left) and hyperbolic (right) geometry. For the first six branches, the value of $n$ is indicated by red numbers. **b** Same data for a Euclidean $\{3, 6\}$ (left) and hyperbolic $\{3, 7\}$ (right) tessellation, each with 85 sites. For the hyperbolic lattice, we additionally show the experimental results (orange squares) obtained from the electric circuit.

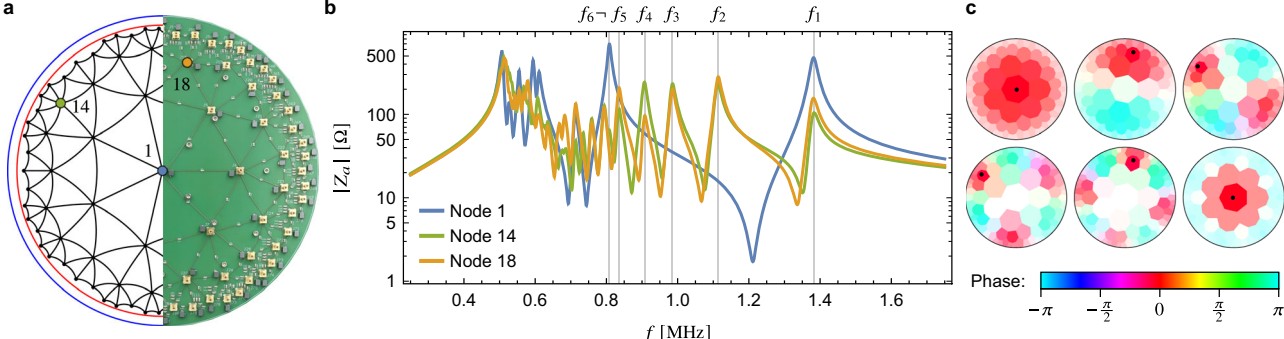

**Fig. 3 | Experimental data. a** Schematic of hyperbolic tessellation (left half) with the unit circle in blue and the circle with radius $r_0 = 0.94$ in red, and photograph of the electric circuit (right half). **b** Measurement of impedance to ground $Z_a$ of the circuit at node $a$ as a function of input frequency $f$ for different nodes (see inset legend and panel **a** for an identification of the nodes). Each impedance peak indicates an eigenmode at that corresponding frequency, which can be excited at the corresponding input node. The highest six frequencies are indicated by vertical gray lines and the corresponding eigenmodes are shown in **c**. **c** Measurement of the voltage profile of the first six eigenmodes (only one mode is shown for each pair of degenerate modes). The saturation encodes the magnitude as a fraction of the voltage (white denotes 0 and full saturation 1) at the input node (black dots), and the color encodes the phase relative to the reference voltage (see legend).

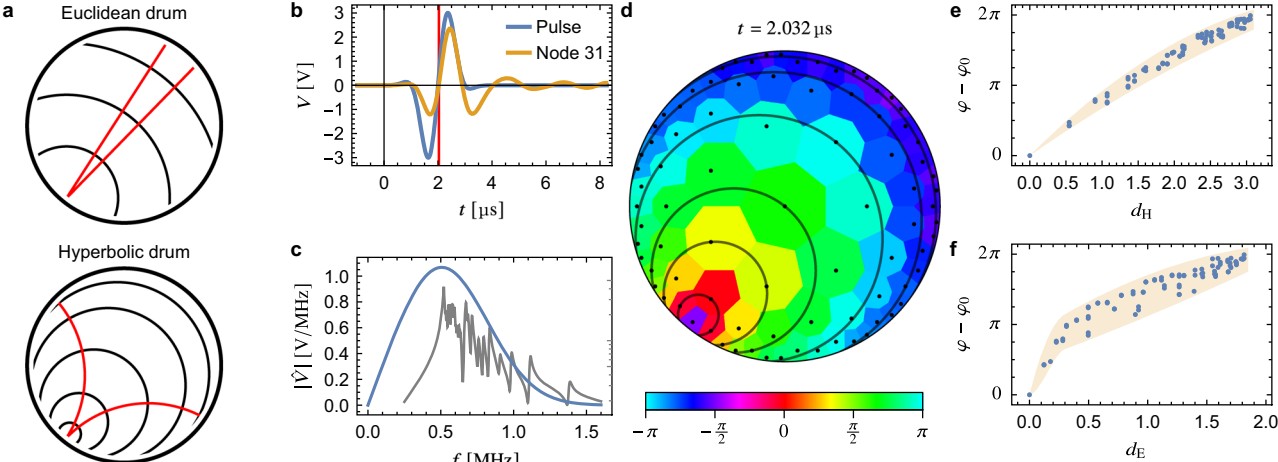

**Fig. 4 | Time-resolved measurement. a** Schematic illustration of the wave propagation after exciting a Euclidean (top) and hyperbolic (bottom) drum with a short and spatially localized pulse. The waves travel along geodesics originating from the source (red lines) and wave fronts at different times are given by concentric circles perpendicular to the geodesics. Several equidistant circles with radii 0.5, 1, … (in the appropriate metric) are shown (black circles) for both cases, illustrating distances $d_E$ and $d_H$ to the source. **b** Broadband excitation pulse (blue) which is fed as a current pulse into node 31 at the boundary, and the voltage response measured at the same node (orange). The time corresponding to the instantaneous phases in **d**–**f** is marked by a red vertical line. **c** Frequency spectrum (blue) of the excitation pulse shown in **b**, demonstrating the wide range of frequencies contained in the pulse by comparison to the impedance to ground $|Z_{31}|$ (gray; shown on a logarithmic scale on the right axis from 20 to 500 Ω). **d** Instantaneous phases of the pulse propagating on the hyperbolic drum (see legend) at time $t = 2.032$ μs. The nodes are indicated by black dots, and concentric hyperbolic circles with center at node 31 are shown in black to illustrate the hyperbolic metric. **e** Difference of the instantaneous phase $\varphi$ at each node to the one at the source of the signal (node 31) $\varphi_0$ vs. the hyperbolic distance $d_H$ to the source. **f** Difference of the instantaneous phase $\varphi$ at each node to the one at the source of the signal (node 31) $\varphi_0$ vs. the Euclidean distance $d_E$ to the the source. The shaded region in **e**, **f** indicates the approximate spread of the instantaneous phase as a function of $d_H$ and $d_E$, respectively.

## Implementation in an electric circuit

In our experiments, the tessellation is realized as an electric circuit network (right half of Fig. 3a) with a node at each site. Nodes are coupled capacitively among each other and inductively to ground. The boundary conditions are implemented by additional capacitive coupling of the nodes in the outermost shell to ground. Effectively, this corresponds to adding one more shell with all nodes shorted to ground, i.e., it represents the lattice equivalent of the Dirichlet boundary conditions. A generic electric circuit network is described by Kirchoff's law:

$$I_a = \sum_b J_{ab}(\omega) V_b,\qquad (3)$$

where $I_a$ and $V_a$ are the input current and voltage amplitude (for angular frequency $\omega$) at node $a$, respectively. The matrix $J(\omega)$ is called[29] the grounded circuit Laplacian, and generally depends on $\omega$. In the continuum limit, the input current $I$ at some position is related to the divergence of the current density $\mathbf{j}$ via $I = \nabla \cdot \mathbf{j}$, with $\mathbf{j} = \sigma \mathbf{E} = \sigma \nabla V$, $\sigma$ the conductivity, $\mathbf{E}$ the electric field due to an applied voltage $V$, and $\nabla$ the del operator (for brevity, we dropped the subscript g indicating the geometry). Hence, $I = \nabla \cdot (\sigma \nabla V) = \sigma \Delta V$, establishing the interpretation of $J$ as the restriction of the continuum Laplace operator to the grounded circuit. The impedance to ground of node $a$, $Z_a(\omega) = V_a/I_a$, is fully determined by $J$ and its resonances correspond to eigenmodes of $J$ with eigenvalues $\lambda \propto 1/\omega^2$ (see Methods). Note that this relationship could be changed to $\lambda \propto \omega^2$ by exchanging the roles of capacitors and inductors in implementing the connections between the nodes resp. to the ground.

Three types of experiments are performed. First, an impedance analyzer is used to measure $Z_a$ as a function of frequency $f = \omega/2\pi$ for each node $a$. The data for three input nodes are shown in Fig. 3b. Second, these eigenmodes are resonantly excited and their voltage profile is measured using lock-in amplifiers. For the modes at the highest six frequencies, both magnitude (relative to the voltage at the input node) and phase (relative to a reference signal) are shown in

Fig. 3c. In the final experiment, the circuit is stimulated by the broadband voltage pulse shown in Fig. 4b fed into the circuit as a current pulse at a node close to the boundary. Subsequently, the voltage is measured as a function of time at each node. We observe the pulse to propagate in the Poincaré disk (the full time dependence is shown in Supplementary Movie 1 and discussed in Supplementary Note 6). A snapshot of the instantaneous phase profile (obtained via a Hilbert transform) is shown in Fig. 4d, which visualizes the propagation of the pulse.

## Evaluation of the experimental data

We proceed with discussing the results of these three measurements. Comparing the impedance of input node 1 (blue curve) to nodes 14 and 18, see Fig. 3b, we clearly observe the spectral reordering discussed in the previous section: there are four additional peaks for input node 14 and 18 located between the two highest-frequency peaks for input node 1. This implies that the second $\ell = 0$ mode (i.e., the first mode with $n > 0$) is the sixth eigenmode. The explicit values of $\ell$ and $n$ for specific modes can be deduced from the voltage profiles of the eigenmodes obtained in the second experiment, see Fig. 3c.

We further plot (orange squares in Fig. 2b) the extracted dispersion of the Laplacian frequencies $\lambda_H^{n\ell}$ with the angular momentum $|\ell|$, obtained by a circular Fourier transform of the measured signal. We observe an almost perfect match with the theoretically predicted values (blue dots in Fig. 2b) for the first few measured modes. However, higher modes are increasingly difficult to excite and detect, due to the finite resolution in frequency and space. We remark that the boundary sites of the present experimental realization of a hyperbolic lattice could be used to probe holographic dualities. For each eigenmode of the system, only its angular distribution on the boundary is important (cf. the angular momentum dispersion in Fig. 2b), yielding a novel and universal one-dimensional physical system on the boundary. We leave a detailed examination of these intriguing edge modes to future studies.

Finally, we discuss the time-resolved measurements. We excite the densest region of the frequency spectrum (Fig. 3b) using a current

pulse (Fig. 4b) of mean frequency 500 kHz (Fig. 4c). By exciting a large number of modes, we approximate the continuum response. The propagation of the pulse through the circuit network leads to the profile of instantaneous phases depicted in Fig. 4d, where the phase fronts can be easily identified by the positions of equal instantaneous phase. Since the connectivity of the nodes implements the metric of the Poincaré disk, these phase fronts form concentric hyperbolic circles, highlighted by black circles in Fig. 4d. This agrees with the theoretical expectation that the signal emanates from the excited node along geodesics, which are the generalization of straight lines in curved space (red lines in Fig. 4a).

Wave fronts are perpendicular to these geodesics and thus constitute concentric circles (black circles in Fig. 4a) up to corrections due to boundary reflections. In Fig. 4d–f, we have chosen an early time during the excitation such that contributions from such reflections do not have a significant impact on the measured phases. Finally, when plotting the phase vs. hyperbolic ($d_H$) and Euclidean ($d_E$) distance in Fig. 4e, f, respectively, we observe that the correlation of the phase with $d_H$ is stronger than with $d_E$. This manifests that the propagation of the signal indeed follows hyperbolic rather than Euclidean geodesics, thereby verifying that the system realizes the hyperbolic rather than Euclidean metric.

## Discussion

We have experimentally simulated the negatively curved hyperbolic plane, as evidenced both in the spectral ordering of the Laplace operator and in the signal propagation along curved geodesics. With an implementation encompassing only 85 lattice sites, we have readily observed an excellent approximation of the hyperbolic plane; at the same time, no technical constraint hinders significantly enlarging the number of sites in future applications. In particular, using existing chip manufacturing technology and commercially available components, electric circuits representing lattices with ~$10^4$ sites should be within reach. In combination with the presented results, the efficient fabricability and high scalability of electric circuits elevates them into a versatile platform for emulating classical hyperbolic models, with several advantages over the previously considered methods[16,24].

First, electric circuits provide easy means for embedding hyperbolic lattices on a flat physical geometry, while allowing for unconnected wire crossings. Such flexibility could be utilized to include coupling beyond nearest neighbors and to implement the plethora of other hyperbolic tessellations[22,23]. In particular, going beyond the presented emulation of the Laplace operator in a negatively curved space, the platform allows to emulate much more complex tight-binding models. These could, for example, be used to test the recently emerging concepts of hyperbolic band theory[12–14], hyperbolic crystallography[15], and hyperbolic topological insulators[19,20]. Electric circuits also excel at providing time- and spatially resolved access to the individual degrees of freedom.

Furthermore, including non-linear and non-reciprocal elements in the network, such as transistors and diodes, is trivial[36]. This enables experimental investigation of how phenomena like topological insulators[19,20,27,28], the non-Hermitian skin effect[37,38], further non-Hermitian topological systems[39] or non-linear topological systems[40–42] interact with with the negative curvature underlying hyperbolic lattices. Given their large scalability, electric circuits could be manufactured with the goal to experimentally study non-linear dynamics of systems with sizes that are unwieldy for numerical simulations. Staying instead within the linear regime, there is a relationship between particles moving freely on geodesics of negatively curved space and deterministic chaos, as illustrated by the Hadamard system[43]. In combination with our experimental verification of the signal propagation along the geodesics, this relationship designates electric circuits a promising experimental platform to investigate classical models of chaos.

Crucially, our work demonstrates the experimental viability of two methods for verifying the hyperbolic nature, i.e., the negative curvature, of the simulated model, which is an important step toward realizing more complicated models. The two methods rely on approximating the Laplace-Beltrami operator using a simple nearest-neighbor tight-binding model and then observing (1) a reordering of eigenmodes compared to flat space, or (2) the propagation of a pulse along hyperbolic geodesics. These methods are, at least in principle, transferable to other platforms, even though it may generally be more challenging to experimentally access the necessary (spatially or time-resolved) information. However, the first method can be applied in a minimal fashion that requires access to significantly less experimental data. As we show in Fig. 3b, it is sufficient to measure the response (here the impedance to ground) at two vertices, one at the origin and one away from it, in order to distinguish $\ell = 0$ from $\ell \neq 0$ modes and observe the predicted mode reordering. In this respect, note that waveguide resonator circuits, were previously proposed as a platform for realizing hyperbolic models as well[16]. However, no substantial experimental verification of the curvature has been performed so far. Our methods could be used to perform a similar analysis on that platform.

Let us finally remark that while coplanar waveguide resonators have been proposed as a promising platform for implementing quantum hyperbolic matter, it is also conceivable[36] that superconducting qubits could potentially be combined with electric circuits in the future. This suggests another route toward exciting future generalizations of our work to quantum models. We expect such generalizations to inspire a new paradigm for designing and measuring holographic toy-models and topological or conformal boundary field theories in discrete geometries. In this context, it is worth reminding that theoretical models of hyperbolic quantum systems were proposed[44], which still await experimental implementation, including MERA tensor networks[5,8] and topological quantum memories[45,46]. These efforts have the potential to fundamentally alter our understanding of physics in curved spaces and imply novel views on problems in condensed matter theory, quantum gravity, cosmology, and holography.

## Methods

### Eigenmodes of the Laplace-Beltrami operator

The solutions to Eq. (2) on the disk $\mathcal{D}_{r_0}$ of radius $r_0 < 1$ correspond to the eigenmodes of a drum of radius $r_0$ in the corresponding geometry. They can be conveniently expressed using special functions. Going to polar coordinates $(r, \theta)$, one finds (cf. Supplementary Note 1) for the Euclidean metric:

$$u_E^{n\ell}(r, \theta) = \mathcal{J}_\ell(k_{n\ell}r)e^{i\ell\theta}, \tag{4}$$

where $\mathcal{J}_\ell(z)$ are the Bessel functions of the first kind and $k_{n\ell}$ is the $(n+1)$th zero of $k \mapsto \mathcal{J}_\ell(kr_0)$. From the angular part of the solution it follows that $\ell$ can be interpreted as the angular momentum. Furthermore, $k_{n\ell} = \frac{z_{\ell,n+1}}{r_0}$, where $z_{\ell,n}$ is the $n$th zero of $\mathcal{J}_\ell(z)$. The radial zeroes $r_m$ of $u_E^{n\ell}(r, \theta)$ are then given by:

$$r_m = \frac{z_{\ell,m}}{k_{n\ell}} = r_0 \frac{z_{\ell,m}}{z_{\ell,n+1}}, \tag{5}$$

such that $m = 1, 2, \ldots, n$ for the non-trivial zeroes $r_m < r_0$. Thus, $u_E^{n\ell}$ has exactly $n$ non-trivial radial zeroes.

For the hyperbolic metric, on the other hand, one finds (cf. Supplementary Note 1):

$$u_H^{n\ell}(r, \theta) = P_{\frac{1}{2}(-1+ik_{n\ell})}^\ell \left(\frac{1+r^2}{1-r^2}\right)e^{i\ell\theta} \tag{6}$$

with $P_\lambda^\ell(z)$ the associated Legendre functions and $k_{n\ell}$ the $(n+1)$th zero of $k \mapsto P_{\frac{1}{2}(-1+ik)}^\ell\left(\frac{1+r_0^2}{1-r_0^2}\right)$. Again we can interpret $\ell$ as the angular momentum and $n$ as the number of radial zeroes of $u_H^{n\ell}$.

## Lattice regularization

The graph Laplacian of a simple (i.e., undirected) graph is given by:

$$Q = A - D, \qquad (7)$$

where $D$ is the degree matrix (the diagonal matrix containing the number of adjacent sites for each site as entries) and $A$ the adjacency matrix ($A_{ab} = 1$ if sites $a$ and $b$ are adjacent and zero otherwise). Assuming the graph represents a lattice regularization of a continuum, then any function $u(x, y)$ induces a function on the lattice, via $a \mapsto u(x_a, y_a) =: u_a$, and the action of the Laplacian matrix, $\sum_b Q_{ab} u_b$, can be expressed in terms of the continuum Laplace-Beltrami operator, e.g., following the steps outlined in ref. 17.

Tessellations of the Euclidean or hyperbolic plane constitute a lattice regularization of the continuum[17], and the boundaries of the tiles (i.e., vertices and edges) can be interpreted as forming a graph. If only a finite segment of the plane is tiled, the tessellation has a boundary, which corresponds to vertices of the graph that are attached to fewer edges than the bulk vertices. Naturally, this is reflected both in the adjacency matrix $A$ as well as in the degree matrix $D$. However, if we impose Dirichlet boundary conditions for $u(x, y)$ as we do in the main text, then $u$ vanishes on the boundary sites, which allows us to drop them from the matrix description. Consequently, we are left only with the bulk part of $Q$. For a Euclidean $\{3, 6\}$ tessellation, we find (cf. Supplementary Note 2):

$$\sum_b Q_{ab} u_b = \frac{3}{2} d^2 \Delta_E u_a + \mathcal{O}(d^3), \qquad (8)$$

where $d$ is the distance between sites. For the hyperbolic tessellation $\{3, 7\}$, on the other hand, we find (cf. Supplementary Note 2):

$$\sum_b Q_{ab} u_b = \frac{7}{4} h^2 \Delta_H u_a + \mathcal{O}(h^3), \qquad (9)$$

where $h = \tanh(d_0) = 0.496\,970$, and $d_0 = 0.545\,275$ is the hyperbolic distance between two neighboring sites in the Poincaré disk representation. For both tessellations, the leading contribution is the Laplace-Beltrami operator for the appropriate metric, such that eigenstates of $Q$ correspond to $u_g^{n\ell}$ from Eq. (2) and the eigenvalues are proportional to $\lambda_g^{n\ell}$ (up to higher-order corrections).

## Extraction of angular momentum dispersion

The angular momentum dispersion, $\lambda_g^{n\ell}$ vs. $|\ell|$, shown in Fig. 2b is extracted from the spectrum and eigenstates of the graph Laplacian using Fourier analysis on shells of the graph, i.e., sites that have approximately the same distance from the disk center and form a circle. A shell can therefore be considered as a one-dimensional system with periodic boundary conditions with the polar angle taking the role of position. For each eigenvector $u$, its components on one of the shells, therefore, define a periodic function $u_{shell}(\theta)$ defined at discrete $\theta$. By first interpolating $u_{shell}(\theta)$ and then performing a discrete Fourier transform on regular samples, we determine the dominant Fourier component which is interpreted as the angular momentum $|\ell|$ of $u$. For the eigenstates shown in Fig. 2b it is sufficient to consider the outermost shell, but for higher eigenstates, considering additional shells can improve the results.

## Theoretical description of electric circuit

In our circuit network, nodes are coupled with capacitance $C$, each node is coupled to ground via an inductance $L$ and the boundary conditions are implemented by adding additional capacitive couplings to ground such that each node is capacitively coupled to seven other components. The grounded circuit Laplacian is then given by the graph Laplacian $Q$ of the underlying (bulk) lattice and a contribution from the inductive grounding (neglecting resistances and other parasitic effects):

$$J(\omega) = -i\omega CQ + \frac{1}{i\omega L}\mathbb{1}. \qquad (10)$$

The spectral decomposition is therefore given by the eigenstates $\psi^\beta$ and eigenvalues $q^\beta$ of the Laplacian matrix, $-Q\psi^\beta = q^\beta \psi^\beta$, with eigenvalues:

$$j^\beta(\omega) = \frac{1 - q^\beta \omega^2 LC}{i\omega L}. \qquad (11)$$

The eigenmode index can be decomposed into the principal and orbital index, $\beta = (n, \ell)$, to match the analytic solution in the continuum.

The inverse of $J$ is called the circuit Green function and can be obtained by expanding $J$ into eigenmodes (here we assume that $J$ is Hermitian and the circuit grounded, as is the case for our circuit) $J(\omega) = \sum_\beta j^\beta(\omega) \psi^\beta \psi^{\beta\dagger}$; then:

$$G(\omega) = \sum_\beta \frac{1}{j^\beta(\omega)} \psi^\beta \psi^{\beta\dagger}. \qquad (12)$$

Assuming current fed into node $a$, i.e., $I_a = \sum_c I \delta_{ca}$, the impedance of that node to ground can be written in terms of the eigenmodes of $J$:

$$Z_a(\omega) = G_{aa}(\omega) = \sum_\beta \frac{1}{j^\beta(\omega)} |\psi_a^\beta|^2, \qquad (13)$$

and the stationary voltage response, i.e., after equilibration, at some other node $b$ is given by:

$$V_b = G_{ba}(\omega) I_a = \sum_\beta \frac{1}{j^\beta(\omega)} \psi_b^\beta \psi_a^{\beta*}. \qquad (14)$$

We observe that in both cases the result is a superposition of eigenmodes of $J$ with the weight proportional to $1/j^\beta(\omega)$, which has a resonance at:

$$\omega^\beta = \frac{1}{\sqrt{LCq^\beta}}. \qquad (15)$$

By combining this result with Eq. (9) for the bulk-to-lattice correspondence, it follows that a resonance of $Z_a$ at frequency $f^\beta = \omega^\beta/(2\pi)$ corresponds to an eigenmode of the hyperbolic drum with eigenvalue:

$$\lambda^\beta = \frac{1}{7\pi^2 h^2 LC} \frac{1}{\left(f^\beta\right)^2}. \qquad (16)$$

This results in a spectral reversal where the lowest-frequency (small $\lambda$) eigenmodes of the Laplace-Beltrami operator correspond to the highest-frequency (large $f$) oscillations of the electric circuit. Equation (16) is used to plot the experimental data in Fig. 2. If the circuit is probed at one of the resonance frequencies, $\omega^\beta$, then the dominant contribution to $V_b$ is:

$$V_b \approx \frac{1}{j^\beta(\omega^\beta)} \psi_b^\beta \psi_a^{\beta*} = \frac{\psi_b^\beta}{\psi_a^\beta} V_a, \qquad (17)$$

where $j^\beta$ does not diverge in practice due to the presence of small resistive terms (see Supplementary Note 4 for a discussion of the impact of parasitic resistances and Supplementary Note 5 for an extended analysis of measured eigenmodes). This implies that the voltage profile encodes the eigenvectors $\psi_b^\beta$.

## Electric circuit parameters

The capacitances of the electric circuit are implemented by ceramic capacitors with $C = 1$ nF and 1% tolerance, the inductances as power inductors with $L = 10$ μH, 5% tolerance and a minimal quality factor of 40 at 1 MHz. Nodes on the boundary have additional capacitors $C$ to ground such that each node is connected to seven identical capacitors in total. Finally, each node is made accessible for in- and ouput via SMB connectors.

## Measurement details

The impedance measurements were performed in a two-terminal measurement configuration using a Zurich Instruments MFIA 5 MHz impedance analyzer. A short/open compensation routine was used to remove the residual impedance and stray capacitance of the test fixture. The impedance of all 85 circuit nodes has been recorded for frequencies in the range from 250 KHz to 1.75 MHz. To exclude transmission line effects in the measurement, the maximum cable length was restricted to be below 1.8 m.

For the measurement of the voltage profiles of the eigenmodes, a reference voltage signal and phase sensitive detection is needed. This was achieved by using three Zurich Instrument MFIA 5 MHz impedance analyzers as lock-in amplifiers synchronized in frequency and phase. Each mode was excited by a current signal of the corresponding frequency fed into the node with the highest impedance peak at that frequency. The current signal was obtained by applying the sinusoidal reference voltage signal with fixed peak-to-peak voltage of 1 V produced by one of the lock-in amplifiers to a shunt resistor of 12 Ω. The other two lock-in amplifiers were used to measure the voltages of the different nodes. All voltage signals demodulated with the reference signal were filtered with a digital low-pass filter of eighth order and a cutoff frequency of $f_{-3\,\text{dB}} = 0.7829$ Hz. The readout of the real and imaginary part of the voltage took place after at least 16 filter time constants which corresponds to at least 99% settling of the low-pass filters in a step response.

The time-resolved measurements were carried out by seven Picoscope 4824, which are eight channel USB oscilloscopes with 20 MHz bandwidth and 12 bit resolution. In the experiment, the circuit was stimulated at node 31 by the broadband pulse:

$$V(t) = V_0 \sin(2\pi f t) e^{-\frac{1}{2}(4(ft-1))^2} \quad (18)$$

with $V_0 = 4.3$ V and $f = 500$ kHz. The pulse is generated by a 50 Ω function generator and the output current was fed directly into the input node. Since the oscilloscopes do not provide a separate trigger channel, one channel of each instrument was fed with a rectangular pulse synchronized with the excitation pulse to trigger the instruments. They used an edge trigger at 1 V in rapid trigger mode and sampled with 40 MS/s, i.e., every 25 ns. Assuming equal behavior of the circuit under repeated stimulation, which was verified during the measurement process by repeating the process described below ten times, the measurement was performed in two steps. First, the seven oscilloscopes were used to measure the voltage at nodes 1 through 49 (including the input node 31), then, in the second run, the input node and nodes 38 through 85 were measured. Finally, the measured real-valued signals $V(t)$ were transformed into complex-valued ones using the Hilbert transform, therefore giving access to the instantaneous phase as the argument of the complex-valued signal:

$$v(t) = V(t) + \frac{i}{\pi}\,\text{p.v.}\int_{-\infty}^{\infty} d\tau\, \frac{V(\tau)}{t - \tau}. \quad (19)$$

## Data availability

All the data (both experimental data and data obtained numerically) used to arrive at the conclusions presented in this work are publicly available in the following data repository: https://doi.org/10.3929/ethz-b-000503548.

## Code availability

All the Wolfram Language code used to generate and/or analyze the data and arrive at the conclusions presented in this work is publicly available in the form of annotated Mathematica notebooks in the following data repository: https://doi.org/10.3929/ethz-b-000503548.

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

## Acknowledgements

P.M.L. and T.B. were supported by the Ambizione grant No. 185806 by the Swiss National Science Foundation. T.N. acknowledges support from the European Research Council (ERC) under the European Union's Horizon 2020 research and innovation program (ERC-StG-Neupert-757867-PARATOP). The work in Würzburg is funded by the Deutsche Forschungsgemeinschaft (DFG, German Research Foundation) through Project-ID 258499086—SFB 1170 and through the Würzburg-Dresden Cluster of Excellence on Complexity and Topology in Quantum Matter—*ct.qmat* Project-ID 39085490—EXC 2147. T.He. was supported by a Ph.D. scholarship of the Studienstiftung des deutschen Volkes. I.B. acknowledges support from the University of Alberta startup fund UOFAB Startup Boettcher and Natural Sciences and Engineering Research Council of Canada (NSERC) Discovery Grants RGPIN-2021-02534 and DGECR-2021-00043.

## Author contributions

R.T. initiated the project, and together with T.N. and T.B. led the collaboration. P.M.L., A.S., L.K.U., T.Ho., T.He. and T.B. performed the theoretical analysis of the hyperbolic tessellations. P.M.L., A.S. and A.V. designed the electric circuit. S.I., H.B. and T.K. carried out the measurements, and together with P.M.L. and A.S. analyzed the collected data. P.M.L., A.S., I.B., T.N. and T.B. wrote the manuscript. P.M.L., A.S., L.K.U., T.Ho., T.He., A.V., M.G., C.H.L., S.I., H.B., T.K., I.B., T.N., R.T., and T.B. discussed together and commented on the manuscript.

## Competing interests

The authors declare no competing interests.
