## [Peer Review File · Nature Communications]

Simulating hyperbolic space on a circuit boardREVIEWERS' COMMENTS

Reviewer #1 (Remarks to the Author):

The authors have done an extensive revision of their draft, and elucidated some of the unclear points that were raised in the first review round. I think the revisions are satisfactory and led to a much improved version of the work.

Reviewer #2 (Remarks to the Author):

I think the authors have done a good job at clarifying the novel aspects of the paper in their manuscript. The research is of high-quality and well presented.

While I think the nature of this work is not radically revolutionary (circuit models are a very well understood platform to implement tight-binding models, and hyperbolic tessellations have been discussed elsewhere) I do think they are novel and interesting enough to a broad community to deserve publication in Nature Communications.

Reviewer #3 (Remarks to the Author):

Lenggenhager et al. emulate a lattice in hyperbolic space with an electronic circuit network and demonstrate two methods for verifying that their system captures the physics of negatively curved space. The first of these is the ordering of eigenmodes, which is notably distinct from that of Euclidean space; the second is the propagation of signals along hyperbolic curved geodesics of the lattice. To the best of my knowledge, the work is original. Its conclusions and claims are well-supported, the methodology is sound, the figures are clear, and the text is organized, clear, and well-written.

The authors have done a great job addressing the comments by all reviewers. This work now highlights the potential for classical electric circuits to explore condensed matter phenomena in curved space; it also provides tools for realizing quantum counterparts. Notably, the authors provide a thorough discussion of possible future research directions.